# Peroxisome Proliferator FpPEX11 Is Involved in the Development and Pathogenicity in *Fusarium pseudograminearum*

**DOI:** 10.3390/ijms232012184

**Published:** 2022-10-12

**Authors:** Mingyu Wang, Hao Xu, Chunjie Liu, Yilin Tao, Xiaofeng Wang, Yuancun Liang, Li Zhang, Jinfeng Yu

**Affiliations:** Key Laboratory of Agricultural Microbiology, College of Plant Protection, Shandong Agricultural University, Tai’an 271018, China

**Keywords:** *Fusarium pseudograminearum*, peroxisome, FpPEX11, pathogenicity

## Abstract

Fusarium crown rot (FCR) of wheat, an important soil-borne disease, presents a worsening trend year by year, posing a significant threat to wheat production. *Fusarium pseudograminearum* cv. b was reported to be the dominant pathogen of FCR in China. Peroxisomes are single-membrane organelles in eukaryotes that are involved in many important biochemical metabolic processes, including fatty acid β-oxidation. PEX11 is important proteins in peroxisome proliferation, while less is known in the fungus *F. pseudograminearum*. The functions of FpPEX11a, FpPEX11b, and FpPEX11c in *F. pseudograminearum* were studied using reverse genetics, and the results showed that FpPEX11a and FpPEX11b are involved in the regulation of vegetative growth and asexual reproduction. After deleting *FpPEX11a* and *FpPEX11b*, cell wall integrity was impaired, cellular metabolism processes including active oxygen metabolism and fatty acid β-oxidation were significantly blocked, and the production ability of deoxynivalenol (DON) decreased. In addition, the deletion of genes of *FpPEX11a* and *FpPEX11b* revealed a strongly decreased expression level of peroxisome-proliferation-associated genes and DON-synthesis-related genes. However, deletion of *FpPEX11c* did not significantly affect these metabolic processes. Deletion of the three protein-coding genes resulted in reduced pathogenicity of *F. pseudograminearum*. In summary, *FpPEX11a* and *FpPEX11b* play crucial roles in the growth and development, asexual reproduction, pathogenicity, active oxygen accumulation, and fatty acid utilization in *F. pseudograminearum*.

## 1. Introduction

Fusarium crown rot (FCR) is a common wheat disease. In Australia, where it was first reported, the direct economic loss due to FCR was as high as AUD 56.3 million per year, with a potential loss of up to AUD 160 million. In the United States, FCR causes wheat yields to decrease by 10 million tons [1,2]. In recent years, the damage caused by FCR has gradually worsened in the Huang Huai wheat region of China; for example, the yield loss caused by FCR is up to 30–50% in many wheat-producing areas in the Henan Province. These data indicate that FCR is a serious threat to wheat grain security and should be controlled immediately. FCR is a typical soil-borne disease. It overwinters mainly in the form of mycelium in soil and diseased plant residues, which can be preserved for more than 2 years in general and longer in arid or semi-arid climates. It is mainly transmitted during cultivation, and its hosts mainly include weeds and various gramineous crops, such as wheat, barley, and corn. It generally does not infect dicotyledonous crops [3,4]. *F. pseudograminearum* shows heterothallism, and the sexual state of G*ibberella coronicola* is not easily inducible indoors and is difficult to find in the field [3]. Pathogens generally invade the roots and stem bases of the plants, and the specific infection site depends on the distribution of the pathogen source in the soil. *Fusarium* head blight could also be induced at the flowering stage in warm and humid climates [5]. The environment in the field is the main factor controlling the development of FCR. It has been reported that early sowing can aggravate the disease, and appropriate late sowing can reduce the degree of the disease [6]. Cohesive, low-lying, poorly drained, or highly humid soil can promote disease development. Excessive application of nitrogenous fertilizer in the field also increases the occurrence of FCR [7,8]. However, an appropriate increase in zinc fertilizer can effectively reduce the occurrence of the disease [9], but the molecular mechanism of its pathogenesis has not yet been revealed.

Peroxisomes, also known as microbodies, are dynamic, small, single-membraned organelles [10], and their quantity and activity can be adjusted according to the state of the tissue and organ, as well as nutrition [11]. When needed, the endoplasmic reticulum (ER) can synthesize new peroxisomes, or they can rapidly divide and proliferate from pre-existing peroxisomes. When the external or cellular environment changes and the peroxisomes are no longer needed to perform their function, they respond to environmental changes through pexophagy. In *S. cerevisiae*, peroxisome division involves several processes. First, mature peroxisomes elongate under the action of Pex11. Matrix proteins and proteins involved in cleavage are delivered to the elongated peroxisomes. Then, Dnm1 localizes to the peroxisomal constriction site to initiate the membrane constriction process by hydrolyzing GTP. Finally, cooperating with Fis1, Dnm1, Mdv1, or Caf4, a daughter peroxisome is produced [12,13,14]. Increasing attention has been paid to the pathogenicity of peroxisomes and plant-pathogenic fungi. First, pathogenic fungi require a large amount of energy when infecting hosts, and most of the energy comes from fatty acid β-oxidation metabolism in the peroxisomes. Second, plants infected by pathogenic fungi produce large amounts of reactive oxygen species (ROS). Phytopathogens can infect plants when they respond to ROS [14]. The peroxisomes contain more than 50 enzymes—mainly catalases and oxidases that are beneficial for cell detoxification—and genes related to peroxisome synthesis are closely related to the pathogenicity of plant pathogens. For example, the deletion of the *PEX5* from *Penicillium chrysogenum* affects asexual reproduction and vegetative growth [15], as well as the proliferation of peroxisomes which accompanies the infection [16]. Most eukaryotic cells contain peroxisomes, which play an integral role in a variety of biochemical pathways, including ether phospholipid biosynthesis, fatty acid α-and β-oxidation, bile acid and docosahexaenoic acid synthesis, glyoxylic acid metabolism, amino acid catabolism, polyamine oxidation, and reactive oxygen and nitrogen metabolism (ROS and RNS) [17]. Additionally, mammalian peroxisomes are not only metabolic organelles but signaling platforms for the regulation of various physiological and pathological processes, including inflammation and innate immunity. For instance, Zellweger syndrome (ZS) is an autosomal recessive peroxisome biogenesis disease, and some reports have pointed out that peroxisome abnormalities can directly or indirectly cause age-related diabetes, neurodegenerative diseases, and cancer [18,19,20]. In plants, peroxisomes are involved in embryonic development, photorespiration, host resistance, metabolism of nitrogen and sulfur compounds, synthesis of plant hormones (auxin, jasmonic acid, etc.), and the glyoxylic acid cycle [21,22,23,24,25]. In [1] seeds, peroxisomes participate in the glyoxylic acid cycle, mobilizing the lipids stored in oleaginous seeds to degrade them into energy for germination, and are therefore also known as a glyoxysomes [26].

PEX11 is a peroxisomal membrane protein and the first protein identified to be involved in peroxisome proliferation and membrane extension [27] and is necessary for the polarization and division of peroxisomal membranes. The PEX11 protein belongs to the PEX gene family, with the most members at present. The number of members in this family varies significantly among different species [28]. Five members of the PEX11 protein family were detected in *Arabidopsis thaliana*—PEX11a, PEX11b, PEX11c, PEX11d, and PEX11e. These are divided into three subfamilies, all of which promote peroxisomal proliferation. Each family member plays a specific role in different environmental conditions, and perhaps in different steps of peroxisome proliferation [28,29]. In mammals, like humans and mice, three PEX11-related proteins exist: PEX11α, PEX11β, and PEX11γ [30]. In fungi, the composition of the PEX11 protein family is relatively complex, containing two to three members in most fungi, and up to five members in ascomycetes. For example, *Magnaporthe oryzae* contains three PEX11 family members, and deletion of the *MoPEX11A* gene has the greatest effect on its pathogenicity [31]. In *S. cerevisiae*, the C-terminal ends of PEX11 are similar to those of the homologous proteins, PEX25p (YPL112c) and PEX27 (YOR193w). PEX11 localizes to the peroxisomal membrane and may form homo-oligomers. These proteins form the *S. cerevisiae* PEX11 protein family, whose members are necessary for peroxisome biosynthesis, and play a role in the regulation of their size and number [32,33]. However, PEX25 is necessary for the de novo biosynthesis of peroxisomes, and PEX27 competes with PEX25 to inhibit peroxisomal proliferation [14]. Thus far, most reports are from studies on yeast and mammals, while the functions of PEX11 in *F. pseudograminearum* are largely unclear.

In this study, reverse genetics was used to investigate the functions of FpPEX11a, FpPEX11b, and FpPEX11c proteins. Our results indicate that FpPEX11a and FpPEX11b play important roles in the growth, asexual reproduction, active oxygen metabolism, and fatty acid utilization of *F. pseudograminearum*. However, the function of FpPEX11c could be confirmed by further study in the future. Most importantly, the absence of any FpPEX11 protein could reduce the pathogenicity of *F. pseudograminearum*.

## 2. Results

### 2.1. Identification and Knockout of FpPEX11 in F. pseudograminearum

Local BLASTp sequence alignment was performed using three members of the PEX11 protein family of *M. oryzae* (MGG_08896, MGG_00648, and MGG_05271), and the PEX11 protein (NP_014494) from *S. cerevisiae* against the *F. pseudograminearum* genome database available in NCBI aided identification of the orthologous genes *FPSE-09675*, *FPSE-09643*, and *FPSE-04578*, which we named FpPEX11a, FpPEX11b, and FpPEX11c, respectively. Further analysis showed that this group of genes had the typical domain of the PEX11 protein in SMART, which was similar to the protein structure of homologous genes in *S. cerevisiae*, *M. oryzae*, and *F. graminearum* (Figure 1). According to MEME, evolutionarily conserved motifs were found at the ends of FPSE-09675, FPSE-09643, and FPSE-04578 (Appendix A). Phylogenetic tree analysis was carried out using undetermined PEX11 proteins of *Neurospora crassa*, *Ogataea angusta*, *Penicillium chrysogenum*, *S. cerevisiae*, *M. oryzae*, *Aspergillus oryzae*, A*spergillus fumigatus*, *Homo sapiens*, *Candida albicans*, and *F. pseudograminearum* (Figure 1). When compared to yeast and other eukaryotes, FpPEX11 appeared to be closely related to *F. graminearum* and *M. oryzae*. Therefore, we designated *FpPEX11a*, *FpPEX11b*, and *FpPEX11c* as *PEX11* protein family members.

To determine the function of FpPEX11 in *F. pseudograminearum* based on the principle of homologous recombination, several hygromycin-resistant transformants were obtained by PEG-mediated protoplast transformation (Appendix A). Taking the *FpPEX11a* gene as an example, the transformant was first preliminarily determined using the detection primer (Appendix A), and then the clean knockout mutant strain ∆*FpPEX11a* was identified by Southern blotting (Appendix A). The complementary gene *FpPEX11a* was integrated into the plasmid pFL2 to obtain the complementary strain ∆*FpPEX11a*-C; ∆*FpPEX11b*, ∆*FpPEX11c*, ∆*FpPEX11b*-C, and ∆*FpPEX11c*-C were obtained in the same manner.

### 2.2. FpPEX11 Is Involved in Vegetative Growth and Asexual Reproduction

We compared the vegetative growth of the ∆*FpPEX11a*, ∆*FpPEX11b*, ∆*FpPEX11c*, WT, and complementation strains on PDA and CM media. When compared to WT, the colony diameter of ∆*FpPEX11a* on PDA medium was significantly lower, the color of the colony was darker, the aerial hyphae were sparser and more curved, and the edge of the ∆*FpPEX11a* aerial hyphae collapsed on CM medium, resulting in premature aging. There was no significant difference between ∆*FpPEX11b* colony diameter and that of WT, but the colony color was darker, the aerial hyphae were significantly reduced, the edge of ∆*FpPEX11b* aerial hyphae collapsed on CM, and the hyphae aged prematurely. No significant variation in vegetative growth of ∆*FpPEX11c* and WT was detected (Figure 2). The results indicate that FpPEX11a and FpPEX11b regulate colony morphology and growth rate, which significantly affect the vegetative growth of *F. pseudograminearum*; however, FpPEX11c had little effect on vegetative growth.

After 3 days of incubation, spore production of ∆*FpPEX11a* and ∆*FpPEX11b* growing in CMC media was reduced by 53.28% and 71.04%, respectively, when compared to that of the WT (Table 1). The clustered conidiophore of ∆*FpPEX11a* and ∆*FpPEX11b* were less than those of ∆*FpPEX11c*. In ∆*FpPEX11a* and ∆*FpPEX11b*, abnormal spore morphology with constriction at the tip was noticed after 6 d incubation. There was no significant difference in spore production or morphology of ∆*FpPEX11c* compared to that of WT. When incubated in YEPD medium, the conidium germination of ∆*FpPEX11a* was reduced by 50% at 3 h post-incubation when compared to that of WT, and the germ tubes were shorter than those of WT. Moreover, ∆*FpPEX11b* and ∆*FpPEX11c* were not significantly different from the WT. From these data, it can be seen that the significant decrease in the number of spores of ∆*FpPEX11a* and ∆*FpPEX11b* was probably due to abnormal sporulation structure, and spore deformity further affected their germination efficiency; *∆FpPEX11c* showed no significant change in asexual reproduction ability when compared to WT (Figure 3). Summarily, FpPEX11a, FpPEX11b, and FpPEX11c played different roles in asexual reproduction of *F. pseudograminearum.* FpPEX11a was involved in the production, morphology, and germination of *F. pseudograminearum* spores to regulate asexual reproduction, while FpPEX11b was not involved in the germination of *F. pseudograminearum spores* and *FpPEX11c* had little effect on asexual reproduction.

### 2.3. FpPEX11 Regulates ROS Metabolism

To investigate whether FpPEX11 is involved in the response to oxidative stress, we analyzed the growth of ∆*FpPEX11a* and ∆*FpPEX11b* on CM medium containing 20 mM H_2_O_2_. Under these growth conditions, we observed obvious defects in the mutants when compared to WT, while ∆*FpPEX11c* was as resistant to oxidative stress as WT. The production of cellular ROS in each strain was qualitatively analyzed by NBT staining. As shown in Figure 4C, the edge color of the ∆*FpPEX11a* and ∆*FpPEX11b* colonies was darker than that of the WT, whereas the color change of ∆*FpPEX11c* was not significant. To verify this result, generation of ROS was visualized by using DHE. Compared to the wild type, the fluorescence of ∆*FpPEX11b* was the strongest, followed by ∆*FpPEX11a* and ∆*FpPEX11c* as the weakest, similar to the WT strain (Appendix A). These results fall in line with NBT findings. This indicates that ROS accumulation and metabolism were inhibited in ∆*FpPEX11a* and ∆*FpPEX11b*, but not in ∆*FpPEX11c*. The above results indicate that the absence of FpPEX11a and FpPEX11b reduced the resistance of the strain to oxidative stress, affected the metabolism of ROS, and thereby caused ROS accumulation, ultimately damaging the detoxification function of peroxisomes. Deletion of *FpPEX11c* had little effect on the response of the strain to oxidative stress.

### 2.4. FpPEX11 Is Involved in Lipid Metabolism

Unlike short-chain fatty acids, which act as substrates for mitochondrial β-oxidation, the main substrate for peroxisome oxidation is medium-long-chain fatty acids [34]. The strain was cultured in MM containing C_14_H_28_O_2_, C_16_H_32_O_2_, C_18_H_36_O_2_, and C_22_H_44_O_2_ to explore the effect of FpPEX11 on the β-oxidation function of peroxisomes.

The number of aerial hyphae of mutants was obviously less than that of the WT, and the longer the carbon chain length, the fewer aerial hyphae. When compared to the WT, the relative growth of ∆*FpPEX11a* and ∆*FpPEX11b* were significantly reduced, and only extremely sparse hyphae were observed in ∆*FpPEX11b* (Figure 5A). No significant difference was noticed in the colony diameter extension of ∆*FpPEX11b*; ∆*FpPEX11c* was not significantly different from the WT strain (Figure 5B). These results indicate that FpPEX11a and FpPEX11b can regulate the utilization of medium-, long-, and very-long-chain fatty acids, while FpPEX11c had little effect on the utilization of fatty acids.

### 2.5. FpPEX11 Is Involved in Responding to Cell Membrane and Cell Wall Stresses

Congo red (CR), a cell wall inhibitor, can hinder the normal assembly of cell walls to produce cell wall stress and inhibit fungal growth [35]. SDS destroys the stability of proteins and fungal cell walls. On CM medium supplemented with 0.01% SDS and 0.2% CR, the colony growth of the mutants was stunted, and the inhibition of ∆*FpPEX11a* and ∆*FpPEX11b* was significantly higher than that of the WT. The growth of aerial hyphae was abnormal, and the sensitivity to cell wall inhibitors increased; Δ*FpPEX11c* growth had no obvious defects when compared to WT and the complemented strain Δ*FpPEX11c*-C (Figure 6). These results indicate that knockdown of FpPEX11a and FpPEX11b reduces the resistance of the cell membrane and cell wall to the external stress.

### 2.6. The ∆FpPEX11 Exhibits Abnormal Peroxisome Number

To investigate the effect of *FpPEX11* deletion on peroxisomes, the peroxisomal membrane protein PMP70 was labeled with GFP. A green punctate distribution was displayed in the WT and all mutants, but fluorescence in ∆*FpPEX11a* and ∆*FpPEX11b* was significantly lower, which indicated that the deletion of *FpPEX11a* and *FpPEX11b* led to a decrease in the number of peroxisomes. The fluorescence intensity in ∆*FpPEX11c* increased, and its volume decreased slightly (Figure 7). This also indicates abnormal biochemical processes, such as active oxygen degradation and fatty acid metabolism, in ∆*FpPEX11a* and ∆*FpPEX11b.* In ∆*FpPEX11c*, the volume of peroxisomes decreased, but their number increased, leading to peroxisomes being supplemented with some functions.

### 2.7. FpPEX11 Regulates Expression of Peroxisome-Proliferator-Associated Gene

While further studying the effect of FpPEX11 on the pathogenicity and functional integrity of *F. pseudograminearum* at the expression level, we detected the effects of the deletion of *FpPEX11a*, *FpPEX11b*, or *FpPEX11c* on each other’s expression levels, peroxisomal proliferation, and β-oxidation-related gene expression levels using quantitative PCR and analyzed their association with the phenotype of each strain.

Through analyzing the *FpPEX11* gene expression difference between ∆*FpPEX11* and WT, we found that the expression of *PEX11b* was downregulated and that of *PEX11c* was upregulated by 2.9 times in ∆*FpPEX11a*. The expression levels of *PEX11a* and *PEX11c* were both significantly upregulated in ∆*FpPEX11b*. The expression of both *PEX11a* and *PEX11b* was significantly downregulated in ∆*FpPEX11c* (Figure 8).

The number of peroxisomes increases through growth and division of pre-existing peroxisomes [36] or, when required, peroxisome biogenesis is initiated de novo from the ER [37]. *PEX3* and *PEX19* are key genes involved in the de novo biosynthesis of peroxisomes. PEX3 is responsible for the secretion of peroxisomal membrane proteins from the ER and acts as a docking protein for PEX19 [38,39]. As a membrane protein receptor, PEX19 recognizes and localizes to peroxisomes [40]. The expression of *PEX3* and *PEX19* was upregulated by 3- to 3.9-fold and 1.1- to 4.3-fold in each mutant when compared to the WT (Figure 8C), but the number of peroxisomes in ∆*FpPEX11a* and ∆*FpPEX11b* was not enhanced (Figure 7). The other pathway for peroxisomal proliferation is the division and proliferation of existing peroxisomes, which occurs through polarized extension and membrane constriction and division. FIS1 is a division factor recruited during division [41], and the expression levels of *FIS1* in the mutants were significantly lower than those in the WT (Figure 8D). DNM1 is a dynamin-related protein involved in membrane constriction that promotes membrane division [42]. The expression of *DNM1* was upregulated in each mutant when compared to that in the WT (Figure 8D). The number of peroxisomes was significantly increased, and the volume was decreased in ∆*FpPEX11c* (Figure 7).

Carnitine acetyltransferase plays an important role in fatty acid degradation. PTH2 is a peroxisome carnitine acetyltransferase that catalyzes the generation of acetyl-CoA from acetyl-CoA and facilitates the transport of acetyl-CoA between organelles. The expression of PTH2 was upregulated in each mutant compared to that in the wild type in each mutant (Figure 8B). PIP2 is involved in fatty acid degradation. Induced by the fatty acid metabolism signal from the body, PIP2 binds to Adr1 to initiate expression and promote fatty acid degradation [43]. The expression of PIP2 in each mutant was upregulated when compared to that in the wild type (Figure 8B).

### 2.8. FpPEX11 Is Essential for the Pathogenicity of F. pseudograminearum

To explore the effect of *FpPEX11* knockdown on the pathogenicity of *F. pseudograminearum* in wheat, wheat coleoptiles were inoculated with spore suspensions of each strain. As shown in Figure 9, when compared to WT, the coleoptile infected length for ∆*FpPEX11a* and ∆*FpPEX11b* was reduced by 70% and 83.4%, respectively, while ∆*FpPEX11c* showed no significant difference when compared to the WT. Further inoculation at the blooming stage of wheat in the field was conducted, and spore suspensions of each strain were prepared in advance. After 10 days of investigation, the disease indices of ∆*FpPEX11a*, ∆*FpPEX11b*, and ∆*FpPEX11c* were significantly reduced when compared to those of the WT. These results indicate that FpPEX11a, FpPEX11b, and FpPEX11c are necessary for the pathogenicity of *F. pseudograminearum* (Figure 9).

DON is an important factor in *Fusarium* pathogenicity. In recent years, *F. pseudograminearum* has been found to produce DON. We explored whether the altered pathogenicity change of ∆*FpPEX11* is related to the abnormal DON synthesis. We tested DON yield, and the results are shown in Table 2. When compared to the WT, the yields of ∆*FpPEX11a* and ∆*FpPEX11b* are significantly reduced, and the yield of ∆*FpPEX11c* is even higher than that of the WT. The yield of DON in the culture medium was quantitatively determined using a beacon DON detection kit. The yields of ∆*FpPEX11a* and ∆*FpPEX11b* were 0. The yield of ∆*FpPEX11c* was 121.43 µg/mg, which was not significantly different from that of the WT (133.78 µg/mg), which is consistent with the trend determined by LC-MS/MS.

To further verify this result, we analyzed the expression levels of *TRI5*, *TRI6*, and *TRI10*, which are related to DON synthesis. The results of fluorescence quantitative analysis showed that the expression levels of *TRI5*, *TRI6*, and *TRI10* in ∆*FpPEX11a* and ∆*FpPEX11b* were all downregulated to different degrees, while the expression levels of *TRI5* and *TRI6* in ∆*FpPEX11c* were upregulated to different degrees. There was no significant difference between the expression levels of *TRI10* and WT (Table 2). In summary, these results indicate that FpPEX11a and FpPEX11b are indispensable for the pathogenicity of *F. pseudograminearum* and affect the expression and biosynthesis of genes related to DON synthesis. The role of FpPEX11c in DON synthesis requires further investigation.

## 3. Discussion

In *S. cerevisiae*, the PEX11 protein family, consisting of PEX11, PEX25, and PEX27, has been shown to play a major role in peroxisome proliferation [44], and Huber demonstrated that PEX27 plays an inhibitory role in peroxisome proliferation in *S. cerevisiae* [14]. Deletion of *PEX11* results in a smaller number of larger peroxisomes, whereas overexpression of PEX11 results in larger number of smaller peroxisomes. In *M. oryzae*, one of three PEX11 are required for peroxisomal proliferation [31]. To verify the presence or absence of peroxisome structures in the mutants, we determined the localization in the mycelia of a known peroxisomal membrane protein, PMP70, and the number of peroxisomes was found to be significantly reduced in ∆*FpPEX11a* and ∆*FpPEX11b* when compared with the WT (Figure 7). To compensate for the number of peroxisomes, the expression of the *PEX3* and *PEX19* genes associated with the de novo biosynthesis of peroxisomes was significantly upregulated, and the number and function of peroxisomes in ∆*FpPEX11* did not recover well.

Downregulation of expression of the mitogen *FIS1* involved in peroxisome proliferation and upregulation of *DNM1* expression with PEX11 as the activator protein might be due to the feedback mechanism of the PEX11 protein (Figure 8). These abnormalities in expression levels of PEX11-related genes also reflect changes in the number of peroxisomes in ∆*FpPEX11*. In ∆*FpPEX11c*, the number of peroxisomes increased, and the volume decreased. Expression analysis of FpPEX11 revealed a decreased number of peroxisomes when the expression of *PEX11c* was higher. It has been speculated that FpPEX11c and PEX27 [14] of yeast also play an inhibitory role in the peroxisome proliferation. FpPEX11a and FpPEX11b are also involved in the regulation of peroxisome proliferation. Therefore, the three PEX11 family members in *F. pseudograminearum* are involved in the regulation of peroxisome proliferation.

A variety of metabolic reactions occur in peroxisomes; these include fatty acid β-oxidation, the glyoxylic acid cycle, melanin biosynthesis, and glycerol accumulation, which are related to the growth and development of pathogens. Currently, it has been shown that the pathogenicity of many plant pathogens, including *M. oryzae*, *Colletotrichum gloeosporioides*, *F. graminearum*, *Ustilaginoidea maydis*, and *Phaeosphaeria nodorum,* has inevitable connection with peroxisomes [45]. In this study, the peroxisome metabolism of the PEX11 knockout was diminished, and the fatty acid β-oxidative metabolism of ∆*FpPEX11a* and ∆*FpPEX11b* was distinctly affected. In media containing medium- or long-chain fatty acids, hyphal growth was inhibited (Figure 5), fatty acids could not be normally used, and supply of the metabolite acetyl-CoA was decreased. Fungal cell walls are mainly composed of chitin, β-1,3-glucan, β-1,6-glucan, and mannoproteins. Some studies have suggested that chitin and glucan in fungal cell walls are derived from acetyl-CoA [46]. Deletion of *PEX5*, *PEX6*, and *PEX19* were all sensitive to Congo red and calcofluor-white in *M. oryzae* [47]. In this study, ∆*FpPEX11a* and ∆*FpPEX11b* were sensitive to cell wall inhibitor stress and showed impaired cell wall integrity (Figure 6), which reflects the impaired peroxisome function observed in ∆*FpPEX11a* and ∆*FpPEX11b*. The number of peroxisomes was significantly increased, and their volumes reduced in ∆*FpPEX11c* (Figure 7), hence, no significant change in the β-oxidation function of peroxisomes was observed. In *S. cerevisiae*, acetyl-CoA, a product of β-oxidation, can be transferred to the cytoplasm by carnitine acetyltransferase (CrAT2), or to the mitochondria by CrAT1, where it enters the tricarboxylic acid cycle [48]. The *M. oryzae* gene *PTH2* (encoding CrAT1) plays a major role in acetyl-CoA metastasis. In the *PTH2* deletion mutant, melanin was reduced, invasion was abnormal, and pathogenicity was lost [49]. In ∆*FpPEX11a* and ∆*FpPEX11b*, PTH2 satisfies the need of peroxisome enzyme metabolism by increasing its expression level (Figure 8). Being stimulated by fatty acid signals, PIP2 was upregulated. Fatty acid degradation was still severely impaired in ∆*FpPEX11a* and ∆*FpPEX11b*, especially the degradation of long-chain fatty acids. Therefore, FpPEX11a and FpPEX11b may play a significant role in regulating β-oxidation.

The early defense response of plants to pathogen infection involves the production of ROS to resist pathogen infection, and the ability of the pathogen to remove ROS is essential for successful invasion and colonization despite the host defense. The abundant enzymes in peroxisomes play an important role in regulating active oxygen balance. In yeast, peroxisome defects lead to ROS accumulation [50]. ROS are also produced during the interaction between *Alternaria alternata* and the host, and the destruction of the host ROS scavenging system leads to the reduction or loss of pathogenicity of *A. alternata*. [51]. In this study, it was found that the susceptibility of ∆*FpPEX11a* and ∆*FpPEX11b* to the oxidant H_2_O_2_ was increased, the accumulation of ROS was increased in hyphae, and the removal of ROS was blocked (Figure 4). On the other hand, the abnormal response of *F. pseudograminearum* to ROS may also be associated with its reduced pathogenicity; ∆*FpPEX11c* showed normal ability to respond to oxidative stress (Figure 4). Indoor coleoptile and field wheat inoculation showed that the pathogenicity of ∆*FpPEX11a* and ∆*FpPEX11b* significantly decreased (Figure 9), and the yield of DON decreased (Table 2). Previous studies have shown that fungi can recognize the host environment and induce gene expression for DON biosynthesis during infection, leading to the production of DON. DON has also been detected in tissues inoculated with *F. graminearum* or *F. pseudograminearum* in the stalk portion and not inoculated at the apex [52]. Thus, the effects of the deletion of *FpPEX11a* and *FpPEX11b* on DON contamination in grains and stalks can be continued. In the laboratory conditions, ∆*FpPEX11c* had no significant effect on the pathogenicity when compared with the WT; however, in the field, the pathogenicity of ∆*FpPEX11c* was significantly decreased (Figure 9), which might be due to differences in the environmental factors of the host. The expression levels of *TRI5*, *TRI6*, and *TRI10*, which are related to DON biosynthesis, were significantly downregulated in ∆*FpPEX11a* and ∆*FpPEX11b*, while the expression of *TRI5* was significantly upregulated in the ∆*FpPEX11c*, and the yield of DON was not significantly different from that in WT. Therefore, comprehensive knowledge about the function of FpPEX11c requires further study in the future.

In conclusion, our results indicate that FpPEX11a and FpPEX11b are involved in the regulation of vegetative growth and asexual reproduction of F. pseudograminearum, and act through regulation in the number and functions of peroxisomes, such as via fatty acid β-oxidation and active oxygen metabolism. After deletion of FpPEX11a and FpPEX11b, the pathogenicity and yield of DON in F. pseudograminearum was significantly weakened, and disease progress was hindered (Figure 9). After deletion of FpPEX11c, the number of peroxisomes increased, and their volume decreased (Figure 7). Therefore, the three members of the FpPEX11 family, FpPEX11a, FpPEX11b and FpPEX11c, perform their respective roles, and their mutual balance regulates the number of peroxisomes and metabolic activities. Furthermore, we characterized part of the function of the FpPEX11 family in F. pseudograminearum for the first time, which can aid in the understanding of mechanisms underlying pathogenicity of F. pseudograminearum and other filamentous phytopathogens.

## 4. Materials and Methods

### 4.1. Fungal Strain and Culture Medium

The *F. pseudograminearum* wild-type (WT) strain used in this study was the local isolate Fp3-3, which was reserved at Shandong Agricultural University. Solid potato dextrose agar (PDA; 200 g peeled potato, 20 g dextrose, 15 g agar, and 1000 mL water), complete medium (CM), yeast extract peptone dextrose (YEPD) medium (yeast extract, 10.0 g; peptone, 20.0 g; glucose, 20.0 g; and 1000 mL distilled water), and sporulation media consisting of carboxymethyl cellulose liquid media (CMC) preparations, as well as the culture conditions, were adopted from a previous report [53,54].

### 4.2. Sequence Analysis

FpPEX11a (FPSE-09675) was identified using the BlastP and tBlastN algorithms from *S. cerevisiae* PEX11 (NC_001147.6) of NCBI (https://www.ncbi.nlm.nih.gov/) (accessed on 15 January 2020), and FpPEX11b (FPSE-09643) and FpPEX11c (FPSE_04578) were identified using the BlastP and tBlastN algorithms from *M. oryzae* PEX11 (MGG-00648, MGG-05271). SMART (http://smart.embl-heidelberg.de/) (accessed on 16 January 2020) predicted the domains. Motif prediction was carried out using MEME (https://meme-suite.org/meme/tools/meme) (accessed on 16 January 2020), phylogenetic analysis of the predicted PEX11 protein of *F. pseudograminearum* with those that have been reported in other species was performed using MEGA 7, and the evolutionary tree was constructed by multiple sequence alignment using Clustal W with 1000 neighbor-joining definite tests. FpPEX11 was used to construct a phylogenetic tree with homologous proteins from different species.

### 4.3. Strain Construction

At least 1 kb upstream and downstream fragments for each gene were amplified from genomic DNA, and the hygromycin-phosphotransferase (hph) gene was amplified from the pCB1003 plasmid using the primer pairs zH1-F/R and zH2-F/R and inserted into pCE-Zero to generate vectors pCE-aAH1, pCE-aH2B, pCE-bAH1, pCE-bH2B, pCE-cAH1, and pCE-cH2B. The fusion fragments were transformed into the WT strain to yield the transformants ∆*FpPEX11a*, ∆*FpPEX11b*, and ∆*FpPEX11c* by PEG-mediated protoplast transformation. Potential gene deletion mutants were further confirmed by Southern blot analysis [31].

For the complementation strains, genomic fragments containing full lengths of ORFs and 1.5 kb upstream of *FpPEX11a*, *FpPEX11b,* and *FpPEX11c* were amplified and inserted into pFL2-GRP to generate complementary vectors pFL2-GRP-aDW, pFL2-GRP-bDW, and pFL2-GRP-cDW, which were integrated into mutants. Candidate complementary strains were further determined by PCR.

### 4.4. Vegetative Growth Detection

In order to determine mycelial growth, each strain was cultured on PDA and CM at 25 °C for 3 days, and the colony diameter was measured by the perpendicular cross measurement method [55]. To determine cell wall sensitivity, each strain was cultured on CM containing 0.01% sodium dodecyl sulfate (SDS) and CM containing 0.2% Congo red for 3 days and then measured using the same method.

Utilization of carbon sources was measured in minimal medium (MM) supplemented with various carbon sources instead of sucrose at the following concentrations: 2.5 mM myristic acid (C14), 2.5 mM palmitic acid (C16), 2.5 mM oleic acid (C18), and 2.5 mM erucic acid (C22). Emulsifier NP40 was added to the MM. Colony diameters were measured after three days of incubation at 25 °C [56].

Three 5 mm bacterial cakes were cultured in CMC at a controlled environment of 25 °C and 150 rpm for 3 days to determine the conidia yield and observe the sporulation structure of each strain. Conidia were cultured in YEPD for 3 h, and the germination rate was calculated after microscopic observation.

### 4.5. ROS Detection

The strains were cultured in CM at 25 °C for 4 days to detect ROS by nitro tetrazolium blue chloride (NBT) staining. Each plate was stained with 15 mL of 0.2% (*w*/*v*) NBT solution and incubated in the dark at 25 °C for 45 min. The solution was drained, and the plates were washed with anhydrous ethanol. The plates were incubated for 30 min in the dark at 25 °C before imaging [16]. Generation of ROS was visualized by using DHE: mycelia harvested from YEPD cultures and were washed with phosphate-buffered saline (PBS). The samples were stained with DHE solution at 2.5 μg/mL in PBS and incubated in the dark for 5 min. The DHE stained cells were then examined under the same fluorescence filter set used for RFP.

### 4.6. Fluorescence Microscopy

Plasmids containing GFP-FpPMP70 were transformed into WT and mutants by PEG-mediated protoplast transformation to obtain fluorescently labeled strains. The fluorescence intensity of tagged proteins was observed using a Zeiss LSM880 confocal microscope (Gottingen, Niedersachsen, Germany). Image analysis was performed using ImageJ (http://rsb.info.nih.gov/nih-image/) (accessed on 20 February 2022) and Adobe Photoshop software. Peroxisome quantification was performed on 150 μm hyphae, and statistical data analysis was performed using Statistica 8.0.

### 4.7. Relative Quantitative Real-Time PCR

Total RNA was isolated from the hyphae of the WT, mutant, and complemented strains using TransZol Up (TransGen Biotech, Beijing, China). Relative quantification relates the PCR signal of the mutant transcript in a treatment group to that of another sample, such as the WT. For each sample, FP-β-tubulin was used as an internal control [57]. Quantitative results were calculated and analyzed using the 2^−ΔΔCt^ method [58]. The primers used are listed in the Supplementary Material.

### 4.8. Pathogenicity Test

The inoculum was prepared containing 1 × 10^7^ conidia in 1 mL of sterile water, to which Triton 60 was added at a final concentration of 0.01% for better adherence of the inoculum to susceptible wheat Jimai 22 [59]. After inoculation, humidity was maintained with a sealing film for 48 h, and wheat heads were photographed and assayed [56] 10 days post-inoculation (dpi). Wheat coleoptiles were inoculated with inoculum and examined at 5 dpi and 25 °C [60].

## Figures and Tables

**Figure 1 ijms-23-12184-f001:**
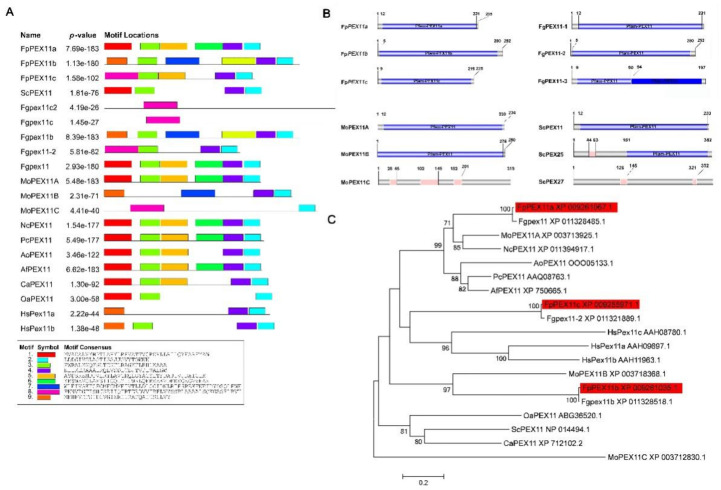
Bioinformatics analysis of PEX11 in *F. pseudograminearum.* (**A**) MEME analysis. (**B**) Domain analysis. (**C**) Phylogenetic tree. (**A**) FpPex11 contains conserved C-terminal motif as revealed by MEME analysis. Sc- *Saccharomyces cerevisiae*; Fg—*Fusarium graminearum*; Mo—*Magnaporthe oryzae*; Nc—*Neurospora crassa*; Pc—*Penicillium chrysogenum*; Ao—*Aspergillus oryzae*; Af—*Aspergillus fumigatus*; Ca—*Candida albicans*; Oa—*Ogataea angusta*; Hs—*Homo sapiens*. (**B**) Prediction domain of PEX11 by SMART analysis. (**C**) Phylogenetic relationship of PEX11 homologs calculated with neighbor-joining method using the MEGA 7.0.

**Figure 2 ijms-23-12184-f002:**
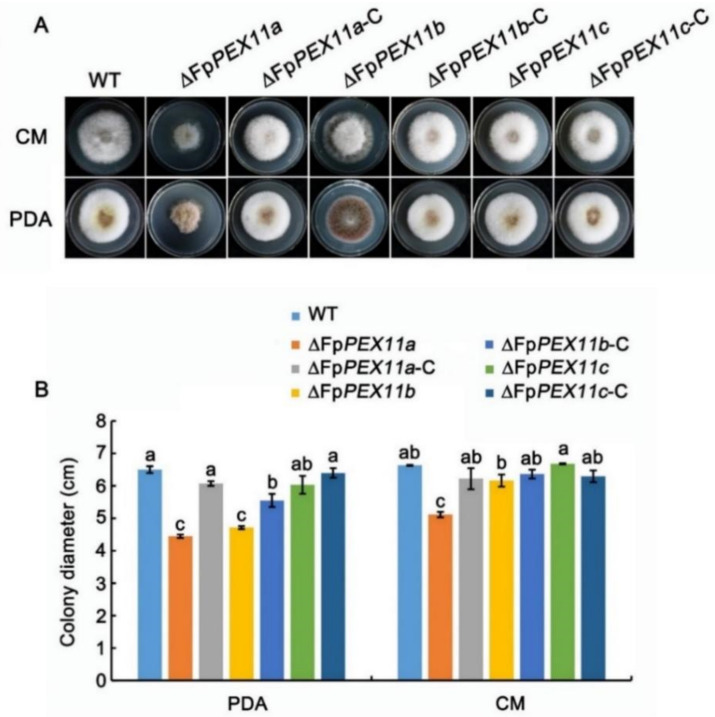
Phenotypic test assay for WT, ΔFpPEX11, and ΔFpPEX11-C strains in *F. pseudograminearum*. (**A**) Mycelial growth of all strains on PDA and CM medium for 3 days. (**B**) Strain colony diameter statistics. Different letter on the bars for each treatment indicate significant difference at *p* < 0.05 by Duncan’s multiple range test.

**Figure 3 ijms-23-12184-f003:**
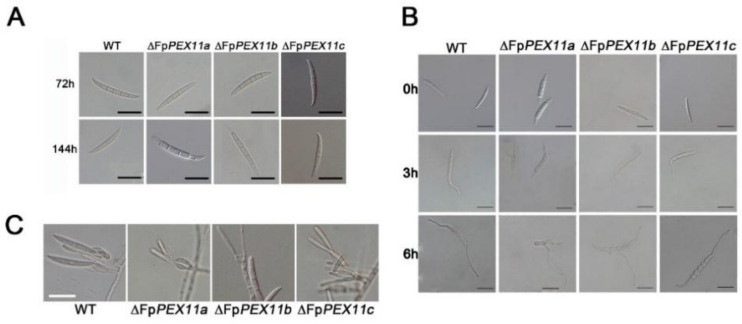
Asexual reproduction assay in WT and ΔFpPEX11 strains in *F. pseudograminearum.* (**A**) Conidia of WT and the mutants incubated in liquid CMC for 3 days. (**B**) Strains were incubated in YEPD culture medium for 3 h or 6 h. Scale bar = 20 μm. (**C**) Conidiogenous structure.

**Figure 4 ijms-23-12184-f004:**
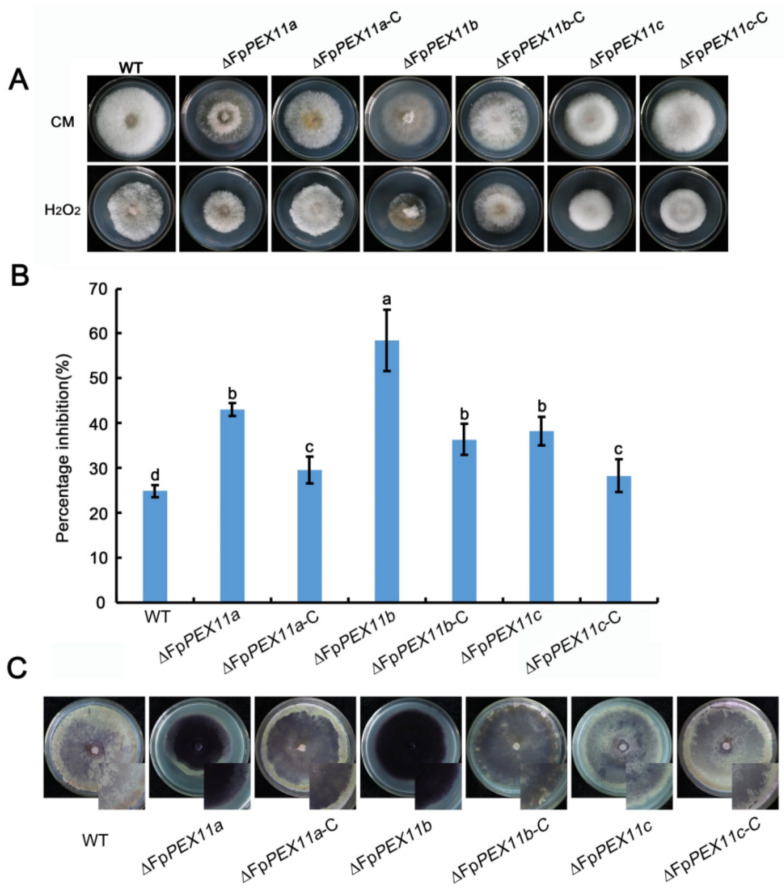
Sensitivity to H_2_O_2_ and ROS accumulation. (**A**) Strains were grown on complete medium (CM) supplemented with H_2_O_2_. (**B**) Mycelial growth inhibition. (**C**) Nitroblue tetrazolium (NBT) staining for ROS production in mycelia of strains. Different letter on the bars for each treatment indicate significant difference at *p* < 0.05 by Duncan’s multiple range test.

**Figure 5 ijms-23-12184-f005:**
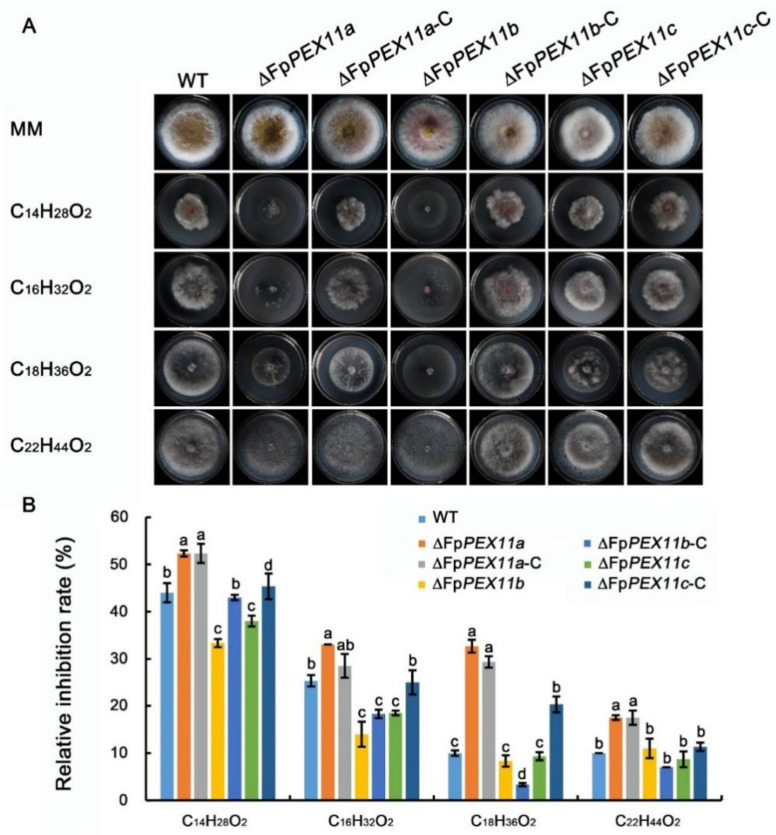
Deletion of strain affected fatty acid β-oxidation. (**A**) Strains were cultured at 25 °C for 3 d on minimal medium (MM) supplemented with different carbon chain length of fatty acids as the sole carbon source. (**B**) Relative mycelial growth of WT and the mutants on various media. Different letter on the bars for each treatment indicate significant difference at *p* < 0.05 by Duncan’s multiple range test.

**Figure 6 ijms-23-12184-f006:**
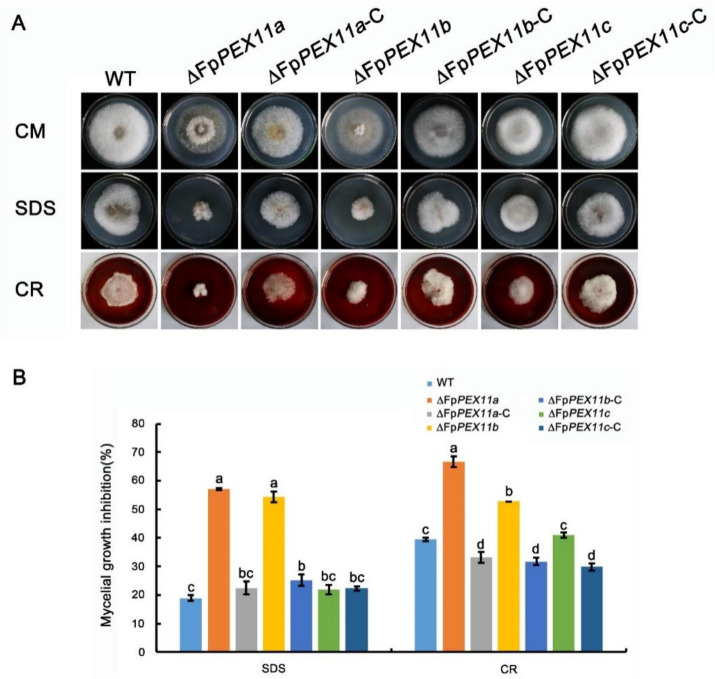
Cell wall integrity of the strains was defective. (**A**) Strains grown on CM supplemented with CR and SDS. Photographs were taken after incubation for 3 days at 25 °C. (**B**) Percent inhibition of strains (%) Inhibition = (diameter of untreated strain-diameter of treated strain)/(diameter of untreated strain) × 100%. Different letters on the bars for each treatment indicate significant difference at *p* < 0.05 by Duncan’s multiple range test.

**Figure 7 ijms-23-12184-f007:**
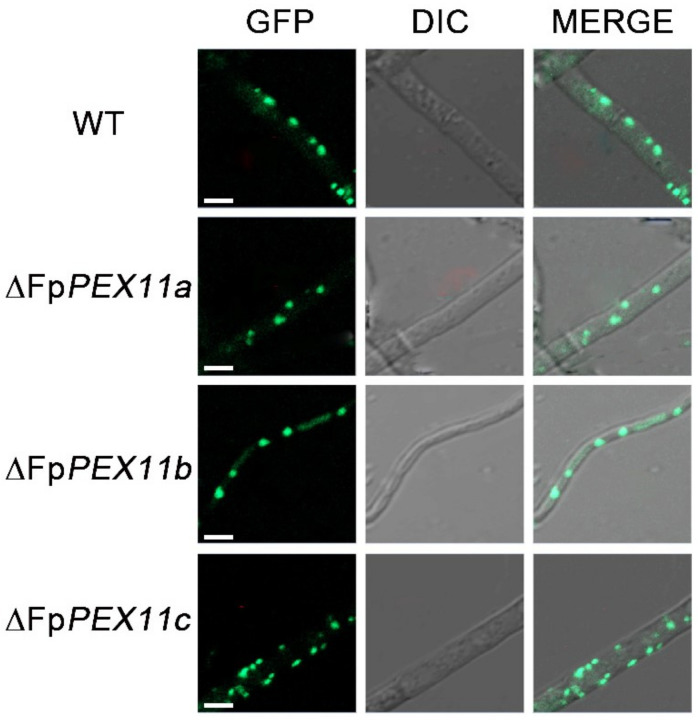
The expression and localization of FpPMP70-GFP in WT and Δ*FpPEX11* mutants using its native promoter by fluorescence microscopy. DIC: differential interference contrast. Scale bar = 5 μm.

**Figure 8 ijms-23-12184-f008:**
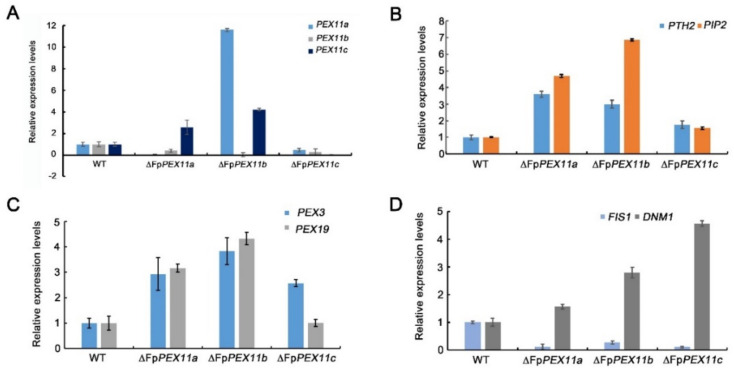
FpPEX11−related gene expression assay results. (**A**) Relative expression levels of mutual PEX11 in the ∆*FpPEX11* mutants. (**B**) Relative expression levels of PTH2 and PIP2 in ∆FpPEX11 mutants. (**C**) Relative expression levels of PEX3 and PEX19 in ∆*FpPEX11* mutants. (**D**) Relative expression levels of FIS1 and DNM1 in ∆*FpPEX11* mutants.

**Figure 9 ijms-23-12184-f009:**
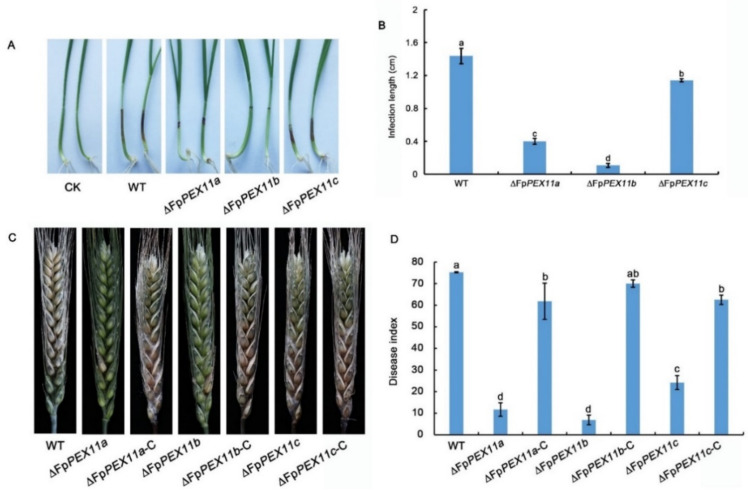
Effect of FpPEX11 knockout on the pathogenicity of *F. pseudograminearum*. (**A**) Infestation of coleoptile 10 d after inoculation. (**B**) Determination of the length of infestation of coleoptile 10 d after inoculation. (**C**) Infestation of wheat ears 10 d after inoculation. (**D**) Disease index statistics of wheat ears 10 d after inoculation. Different letter on the bars for each treatment indicate significant difference at *p* < 0.05 by Duncan’s multiple range test.

**Table 1 ijms-23-12184-t001:** Conidiation and conidial germination for 3 h or 6 h in the WT and Δ*FpPEX11* strains.

Strain	Conidiation (10^6^ Conidia/mL)	Germination (%)	
		3 h	6 h
WT	2.59 ± 0.15 a	51.6 ± 0.75 a	100.00 ± 0.00 a
∆*FpPEX11a*	1.21 ± 0.10 b	27.2 ± 3.32 b	97.00 ± 3.00 a
∆*FpPEX11b*	0.75 ± 0.09 b	49.2 ± 1.16 a	95.00 ± 1.00 a
∆*FpPEX11c*	2.37 ± 0.35 a	52.8 ± 1.59 a	99.00 ± 1.00 a

Different letters for each treatment indicate significant difference at *p* < 0.05 by Duncan’s multiple rang.

**Table 2 ijms-23-12184-t002:** DON production in the WT and ∆FpPEX11 strains.

	DON Production		Relative Expression Levels of *TRI*		
	ELISA (μg/mg)	HPLC (μg/g) *	*TRI5*	*TRI6*	*TRI10*
WT	133.78 ± 18.36 a	14.59 ± 3.47 b	1.00 ± 0.09	1.00 ± 0.22	1.00 ± 0.22
∆*FpPEX11a*	0	0.16 ± 0.01 c	0.11 ± 0.04	0.16 ± 0.06	0.19 ± 0.02
∆*FpPEX11b*	0	0.00763 ± 0 c	0.13 ± 0.02	0.10 ± 0.02	0.15 ± 0.08
∆*FpPEX11c*	121.43 ± 21.65 a	32.05 ± 6.00 a	7.198 ± 0.09	1.82 ± 0.24	0.90 ± 0.21

DON production was analyzed by HPLC–MS/MS (HPLC) and Deoxynivalenol (DON) Rapid Test Kit (ELISA). * Logarithmic transformation of DON production by HPLC and the different letter on the bars for each treatment indicate significant difference at *p* < 0.05 by Duncan’s multiple range test.

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
