# Peer review of "Peroxisome Proliferator FpPEX11 Is Involved in the Development and Pathogenicity in Fusarium pseudograminearum"

_ijms, 2022, doi:10.3390/ijms232012184_

Round 1

Reviewer 1 Report

This manuscript mainly reported the characterization of three genes, FpPEX11a, FpPEX11b, and FpPEX11c, in F. pseudograminearum indicating their involvement in the growth and development, asexual reproduction, pathogenicity, active oxygen accumulation, fatty acid utilization, DON production of F. pseudograminearum. The content itself could be of potential interest as it might expand our understanding of fungal pathogenesis and other biological characteristics of peroxisomes. I have some suggestions showed below:

1.The abstract should be described in more detail that included the main results obtained from these experiments. For example, the functions of FpPEX11 to regulate the expression of peroxisome proliferator associated genes and the production of DON could be included in the abstract.

2.The article format needs to be adjusted according to the requirements of the journal.    The Figures and Tables should be incorporated into to the main text in order.

3.In line 140, i think “the number of aerial hyphae was significantly reduced--” could be changed as “the aerial hyphae were significantly reduced”.

4.In line 159 to 162, this sentence should be more clearly stated since FpPEX11b was not involved in the germination of F. pseudograminearum spores.

5.The figure captions should be examined carefully and described more accurately. For example, in figure 3A, this is not the Perithecia.

6.The parages of 2.1.2 and 2.1.5 could be merged, because they all described the abiotic stress response of the mutants.

7.For qualitatively analysis of the cellular ROS production, i think the mycelium could be examined by microscopy.

8.Some language problems in the text also need to be strengthened.

Author Response

Dear Reviewer:

Thank you very much for your letter and the comments from the referees about our paper submitted to International Journal of Molecular Science. We have learned much from your comments, which are fair, encouraging and constructive. After carefully studying your comments and advice, we have made corresponding changes. Below is our point-by-point response to your comments.

  1. The abstract should be described in more detail that included the main results obtained from these experiments. For example, the functions of FpPEX11 to regulate the expression of peroxisome proliferator associated genes and the production of DON could be included in the abstract.

Response: Thank you for your advice and the related content has been added in the abstract.

  1. The article format needs to be adjusted according to the requirements of the journal.    The Figures and Tables should be incorporated into to the main text in order.

Response: Thank you for your advice and the related content has been added in the manuscript.

  1. In line 140, i think “the number of aerial hyphae was significantly reduced--” could be changed as “the aerial hyphae were significantly reduced”.

Response: Thank you for your advice and the related content has been added in the abstract.

4.In line 159 to 162, this sentence should be more clearly stated since FpPEX11b was not involved in the germination of F. pseudograminearum spores.

Response: Thank you for your advice and the related content has been added in line 179 to 183.

5.The figure captions should be examined carefully and described more accurately. For example, in figure 3A, this is not the Perithecia.

Response: Thank you for your advice and the related content has been added in the figure.

6.The parages of 2.1.2 and 2.1.5 could be merged, because they all described the abiotic stress response of the mutants.

Response: We appreciate your consideration very much.

We believe that detecting the accumulation of ROS can reflect the ROS degradation function of peroxisomes in the samples. Colony extension of the samples was threatened (Figure 2), therefore, the cell wall integrity of the samples was further observed by SDS and CR.

7.For qualitatively analysis of the cellular ROS production, i think the mycelium could be examined by microscopy.

Response: Accumulation of ROS in mycelia was visualized with dihydroethidium (DHE) staining and fluorescence microscopy. We put the figure in Supplementary Materials.

8.Some language problems in the text also need to be strengthened.

Response: Thank you very much for your careful. We have employ Editage (www.editage.com) for English language editing. We hope the quality of English in the article can meet the requirements of you and the magazine

Reviewer 2 Report

The manuscript entitled “Peroxisome Proliferator FpPEX11 is involved in development and pathogenicity in Fusarium pseudograminearum” is a research paper focusing on the establishment of the role of peroxisome proliferators in pathogen infestation. To do that they developed a strong group of mutants, but plant-pathogenic fungus interaction is majorly missed in this study. As this paper talks about Fusarium crown rot disease in wheat, major experimental evidences are required to confirm the role of peroxisome proliferator in the pathogenic response in plants.

1.     The Abstract requires more clarification about the topic

2.     Line no. 18: Change “deleing” to “deleting”

3.     Introduce Fusarium pseudograminearum as a fungus in the abstract

4.     The introduction is so descriptive, it should be more composed. Complete rearrangement of sentences is required. Missing abbreviations. The transition of information is not smooth. It should be flawless.

5.     I would suggest authors to break longer sentences into smaller ones nd add references to support and explain that

6.     The information about Fusarium crown rot and the fungus is broadly missing in the introduction

7.     Introduction also lacks hypothesis

8.     Figure 3A: picture quality should be improved

9.     The line number is not continuous, which is confusing

10.  Remove extra “that” from section 2.1.7, line no:18.

11.  Refer to every figure after mentioning any observation throughout the result section, which is missing.

12.  I would suggest combining result and discussion together or discussion part should also be divided into several sections as result.

13.  The discussion is poorly written. Rearrange Line 05 to 10.

14.  Lack of reasoning and references was observed throughout the discussion section.

15.  The wheat-fungus interaction is not clearly explained. It would be better to know the physiological, biochemical, and cellular changes in plant after mutants and WT infestation.

16.  Conclusion section should be added separately with an overview of future research. It would be good to add a graphical overview of the whole research in a nutshell for better understanding.

Author Response

Dear Reviewer:

Thank you very much for your letter and the comments from the referees about our paper submitted to International Journal of Molecular Science. We have learned much from your comments, which are fair, encouraging and constructive. After carefully studying your comments and advice, we have made corresponding changes. Below is our point-by-point response to your comments.

  1. The Abstract requires more clarification about the topic

Response: Thank you for your advice and the related content has been added in the abstract.

  1. Line no. 18: Change “deleing” to “deleting”

Response: Thank you very much for your careful.

  1. Introduce Fusarium pseudograminearumas a fungus in the abstract.

Response: Thank you for your advice and the related content has been added in the abstract and introduction.

  1. The introduction is so descriptive, it should be more composed. Complete rearrangement of sentences is required. Missing abbreviations. The transition of information is not smooth. It should be flawless.

Response: Thank you for your advice and we have try our best to improve the introduction.

  1. I would suggest authors to break longer sentences into smaller ones nd add references to support and explain that

Response: Thank you for your advice. We agree with you and have made changes in the manuscript.

  1. The information about Fusarium crown rot and the fungus is broadly missing in the introduction

Response: Thank you for your advice and the related content has been added in the introduction.

  1. Introduction also lacks hypothesis

Response: Thank you for your advice and we have try our best to improve the introduction.

  1. Figure 3A: picture quality should be improved

Response: Thank you for your advice and we have try our best to improve the  quality of the pictures.

  1. The line number is not continuous, which is confusing

Response: We added the line number again this time, please check whether it meets the requirements.

  1. Remove extra “that” from section 2.1.7, line no:18.

Response: Thank you very much for your careful.

  1. Refer to every figure after mentioning any observation throughout the result section, which is missing.

Response: Thank you for your advice and the related content has been added in the result.

  1. I would suggest combining result and discussion together or discussion part should also be divided into several sections as result.
  2. The discussion is poorly written. Rearrange Line 05 to 10.

Response: Thank you for your advice and we have rearrange this section in the discussion.

  1. Lack of reasoning and references was observed throughout the discussion section.

Response: References has been added in the discussion.

  1. The wheat-fungus interaction is not clearly explained. It would be better to know the physiological, biochemical, and cellular changes in plant after mutants and WT infestation.

Response: The production and gene expression of virulence factor DON in F. pseudograminearum were detected. Thank you for your new ideas, we're going to absorb.

  1. Conclusion section should be added separately with an overview of future research. It would be good to add a graphical overview of the whole research in a nutshell for better understanding.

Response: Thank you for your advice and  the related content has been added in the discussion.

Round 2

Reviewer 2 Report

Accepted